# Integrative Clinical and DNA Methylation Analyses in a Population-Based Cohort Identifies *CDH17* and *LRP2* as Risk Recurrence Factors in Stage II Colon Cancer

**DOI:** 10.3390/cancers15010158

**Published:** 2022-12-27

**Authors:** Benjamin Tournier, Romain Aucagne, Caroline Truntzer, Cyril Fournier, François Ghiringhelli, Caroline Chapusot, Laurent Martin, Anne Marie Bouvier, Sylvain Manfredi, Valérie Jooste, Mary B. Callanan, Côme Lepage

**Affiliations:** 1Faculty of Health Sciences, University of Burgundy, 21000 Dijon, France; 2Institut National de la Santé et de la Recherche Médicale (INSERM) UMR1231, 21000 Dijon, France; 3Department of Pathology, Dijon University Hospital, 21000 Dijon, France; 4Unit for Innovation in Genetics and Epigenetics in Oncology (IGEO) and CRIGEN (Crispr Functional Genomics), Dijon University Hospital, 21000 Dijon, France; 5Genetics and Immunology Medical Institute (GIMI), 21000 Dijon, France; 6Centre Georges-François Leclerc (CGFL), 21000 Dijon, France; 7Department of Hepato-Gastroenterology and Digestive Oncology, Dijon University Hospital, 21000 Dijon, France

**Keywords:** epigenetics, stage II colon cancer biomarkers, LINE1, oncogene signalling, immune contexture, transforming growth factor beta (TGFβ)

## Abstract

**Simple Summary:**

Stage II colon cancer, although manageable by surgical resection with or without adjuvant chemotherapy, still accounts for 16% of colorectal cancer deaths. Prognostic biomarkers are keys to the risk stratification of patients and the decision to recommend adjuvant chemotherapy, but these are currently lacking. Here, by using a French stage II CC population-based cohort to perform DNA methylation and clinical screens (383 patients), we uncovered a methylation classifier that separates stage II CC into four disease subclasses. Survival analysis revealed two methylation sites localised at the *CDH17* and *LRP2* genes, respectively, to be able to predict risk of cancer recurrence. Hypermethylated *CDH17* conferred high risk of disease recurrence and was associated with over-activity of oncogene signalling pathways, such as *KRAS* and low anti-tumour immune activity. Conversely, hypermethylation of the LRP2 gene identified relatively good prognosis stage II CC tumours characterised by intact DNA repair pathways and active anti-tumour immunity.

**Abstract:**

Stage II colon cancer (CC), although diagnosed early, accounts for 16% of CC deaths. Predictors of recurrence risk could mitigate this but are currently lacking. By using a DNA methylation-based clinical screening in real-world (*n* = 383) and in TCGA-derived cohorts of stage II CC (*n* = 134), we have devised a novel 40 CpG site-based classifier that can segregate stage II CC into four previously undescribed disease sub-classes that are characterised by distinct molecular features, including activation of MYC/E2F-dependant proliferation signatures. By multivariate analyses, hypermethylation of 2 CpG sites at genes *CDH17* and *LRP2*, respectively, was found to independently confer either significantly increased (*CDH17*; *p*-value, 0.0203) or reduced (*LRP2*; *p*-value, 0.0047) risk of CC recurrence. Functional enrichment and immune cell infiltration analyses, on RNAseq data from the TCGA cohort, revealed cases with hypermethylation at *CDH17* to be enriched for *KRAS*, epithelial-mesenchymal transition and inflammatory functions (via IL2/STAT5), associated with infiltration by ‘exhausted’ T cells. By contrast, *LRP2* hypermethylated cases showed enrichment for mTORC1, DNA repair pathways and activated B cell signatures. These findings will be of value for improving personalised care paths and treatment in stage II CC patients.

## 1. Introduction

Colorectal cancer (CRC) is the second most common cancer in Europe, accounting for 8% of all cancer deaths and is the 2nd leading cause of cancer deaths, globally [1]. Around 75% of patients may benefit from surgical resection with curative intent [2,3]. Of these, 27% are diagnosed with stage II disease and 23% at stage III. Nearly 30% of stage II, and over 55% of stage III patients, present a recurrence or a metachronous cancer within five years of initial treatment [2,3,4,5]. Death after recurrence following stage II and stage III CC contributes to 18% and 40%, respectively, of mortality from CC. This high risk of lethal disease recurrence identifies early predictive and postoperative monitoring as crucial goals in CC patient management, specifically in order to detect residual malignant disease or metachronous cancers at a curable stage.

While the management of stage III CC after surgery is well standardised, strategies for stage II CC are less well-defined and no benefit has been observed in this subgroup for treatment with adjuvant chemotherapy with 5-FU [6,7]. Defining early stage II CC with ‘high risk’ features will be of value to guide decisions on adjuvant therapy, and this is still under investigation [8]. Regarding histology and molecularly-driven treatment decision-making, the only widely accepted test is to screen for microsatellite instability (MSI) status, which, when positive, is indicative of mismatch repair deficiency (dMMR). Approximately 20% of stage II CC show dMMR where it is predictive of good prognosis with potentially no benefit from adjuvant chemotherapy by 5-FU [8,9].

Additional molecular markers of interest include the CpG island hypermethylator phenotype (CIMP), which is as an independent prognostic factor in MSS colon cancer [10,11]. More recently, molecular subtyping of CC into discreet biological entities called consensus molecular subgroups (CMS) [12] has emerged and these have been shown to have prognostic value in, at least, advanced CC [9], although tumour heterogeneity can be a confounding factor [9]. The clinical relevance of CMS to stage II CC is not as yet clear, at least in clinical trials, and ‘real-world’ analyses are lacking, thus further impeding progress for clinical management of stage II CC.

In this context, we set up an integrated clinical, histo-molecular, and DNA methylation-based discovery strategy for identification of novel biomarkers of risk of recurrence in stage II CC. We based our strategy on a French registry-based stage II CC population (discovery cohort; 383 patients with cryopreserved tumour biopsies) and a curated TCGA cohort of stage II CC patients for whom DNA methylation and RNAseq data were available (validation cohort; 134). By this approach, combined with recurrence free survival analysis, we uncover novel stage II CC disease entities and identify *CDH17* and *LRP2* as independent predictors of disease recurrence in stage II CC.

## 2. Materials and Methods

### 2.1. Patients, Clinical Data and Tumour Samples

Tumour and control biopsy samples (tumour and adjacent normal mucosa) were obtained from 383 patients resected for a stage II colon cancer between January 1998 and August 2007 selected from the Burgundy Registry of Digestive Cancers, Dijon, France. Nine DNA samples from normal colon mucosa (with three matched to tumour samples) and a pool of tumour DNAs were also analysed as controls. Tissue quality controls and DNA extraction were performed, as previously described (see Appendix A for detailed protocols) [13]. The CPP EST I committee (Comité de Protection des Personnes: Committee for the Protection of Individuals) approved the use of these biological collections. Tissue samples were considered surgical waste in accordance with French ethical laws (L.1211-3 to L.1211-9).

Demographic and clinical data for the study patients were extracted from registry annotations. All registry data were collected and validated according to recommendations from the international agency for research on cancer (https://www.iarc.fr, accessed on 1 January 2022), the European Network of Cancer Registries (https://encr.eu/registries-network, accessed on 1 January 2022), and the FRANCIM network (https://lesdonnees.e-cancer.fr/Informations/Sources/SOURCE-Reseau-FRANCIM, accessed on 1 January 2022).

Tumours occurring between the caecum and the splenic flexure were defined as proximal. Cancer extension at the time of diagnosis was classified according to the 5th edition of the tumour-node-metastasis classification provided by the Union for International Cancer Control [14]. Age was categorised according to the third-party distribution of the population: under 65 years, 65–74, and 75 years and above.

A subgroup of tumours was further analysed by immunohistochemistry for total T cell infiltration by immunohistochemistry by anti-CD3 antibody labelling (see Appendix A).

### 2.2. Characterisation of the CpG Island Methylator Phenotype (CIMP), MSI, Chromosomal Instabilities, Gene Mutations and LINE-1 DNA Methylation Status

The CIMP phenotype was determined using a consensus marker panel (Appendix A) by methylation-specific PCR (193 cases) or by MS-HRM (Methyl-Sensitive PCR—high-resolution-melting curve analysis), for the remaining 190 cases. For 68 samples, results were excluded due to poor quality (see online methods for details of the analysis).

Microsatellite instability was evaluated by PCR analysis covering 13 microsatellite sequences (Appendix A) and PCR product sizing on an ABI 3130XL sequencer (Applied Biosystems (Waltham, MA, USA), Thermo Fisher Scientific (Waltham, MA, USA)), using standard procedures. Samples were classified as « MSI-High » if four or more markers were unstable, «MSI-Low» if one to three markers were unstable and «MSS» if all markers were stable.

Chromosomal instability was evaluated using a QMPSF technique (quantitative multiplex PCR of short fluorescent fragment) that investigates the copy number of eight genomic markers, including *TP53* and *EGFR*, with permission (technique is under patent) [15].

Mutation analysis for codons 12 and 13 (exon 2) of *KRAS*, codon 600 (exon 15) of *BRAF,* and exons 545 and 1047 (exon 9 and 20 respectively) of *PIK3CA* were performed by pyrosequencing (PyroMark Q24, QIAGEN^®^) using custom primers (Appendix A), following the supplier’s recommendations.

LINE-1 methylation status was assessed, as described [13].

### 2.3. DNA Methylation and Clustering Analyses in the Population-Based Stage II CC Patient Cohort

DNA methylation analysis was performed on genomic DNA (2 µg) extracted from fresh frozen tumour biopsies obtained from 383 stage II CC patients, following DNA bisulfite conversion and hybridisation analysis on GoldenGate methylation array (Illumina), using the Methylation Cancer Panel I, according to manufacturer’s instructions. Analyses were performed on academic contract by IntegraGen. The methylation Cancer Panel I investigates a set of 1505 CpG sites, covering 807 cancer-related genes (tumour suppressors, oncogenes, genes involved in DNA repair, cell cycle control, differentiation, apoptosis, X-linked, or imprinted genes) and processed for analysis according to published methods [16] (see Appendix A for detailed protocols).

### 2.4. Reanalysis of DNA Methylation and RNAseq Data from Stage II CC from TCGA; Validation Cohort

Detailed procedures for data processing and reanalysis of DNA Methylation and RNA-seq data from TCGA data sets of stage II CC from TCGA are given in the Appendix A. Briefly, DNA methylation reanalysis was performed in a group of 134 samples with available data [obtained by the Infinium Human Methylation 450K BeadChip assay (Illumina)] by using a set of 62 CpG sites that closely mapped to or nearby the 40 CpG sites of our initial classifier (see Appendix A). A gene-set enrichment analysis (GSEA) pathway analysis [17] was performed by comparing quartile 4 cases (tumours displaying the 25% higher beta values for DNA methylation) to quartile 1, 2, and 3 cases (tumours displaying the 75% lowest beta values) for two CpG sites of interest identified in this study (see above, *CDH17* and *LRP2*). Significantly enriched pathways were retained considering a *p*-value inferior to 0.05 and a false discovery rate (FDR) inferior to 0.25. Immune cell subtype analysis was conducted using the ImmuCellAI web application (http://bioinfo.life.hust.edu.cn/ImmuCellAI#!/ (accessed on 1 January 2022)) [18]. Immune contexture analysis was performed by iAtlas [19] on the (https://isb-cgc.shinyapps.io/iatlas/ (accessed on 1 January 2022)) web application.

### 2.5. Statistical Analysis of Clinical Data

For relapse estimations, the minimal follow-up of patients was set at five years. Postoperative deaths (deaths occurring within six weeks after surgery) were excluded to avoid bias. Univariate Cox survival models were estimated for each of the 40 selected CpG sites (M-values for each CpG site were converted to factors by a quartile approach). Multivariate Cox survival models were estimated only for significant CpG sites (*p*-value inferior or equal to 0.05) identified by the univariate analysis. Survival curves were plotted using the Kaplan-Meier method, and survival estimations were compared using the log-rank test (Appendix A).

## 3. Results

### 3.1. Identification of a Novel DNA Methylation Classifier in Stage II Colon Cancer

In a series of 383 tumour samples from stage II colon cancer patients (recruited from the Burgundy cancer registry, see Table 1 for patient characteristics and Figure 1 for study design), we first performed a DNA methylation screen covering 1505 cancer-linked CpG sites. In view of the heterogeneity of stage II CC, our next step was to use an unsupervised clustering approach on our DNA methylation dataset to probe for the existence of potentially novel disease subtypes. By this approach, we uncovered a set of 40 CpG sites, localising to 40 genes involved in cell signalling, developmental, and immune regulatory pathways that could segregate our 383 stage II CC cases into four distinct clusters (named here, 1 to 4) (Table 2; Figure 2a, left panel and Appendix A). Cluster 1 was the smallest group, identified with just 20 individuals (Figure 2a, left panel). Clusters 2, 3, and 4 were composed of 73, 132, and 158 individuals, respectively (Table 2 and Figure 2a, left panel). We next assessed whether DNA methylation clusters 1 to 4 were associated with known clinical and molecular characteristics in stage II colon cancer (Table 2). Although a comparatively small patient group, cluster 1 CC patients showed hypo-methylation of most of the CpG sites compared to the three other stage II CC patient clusters (Figure 2a, left panel). Quite strikingly, cluster 2 patients appeared enriched in *KRAS* mutant cases (41.1% of cluster 2 patients vs. 30.3% for the whole cohort, *p* = 0.0203) and showed a higher than expected proportion of the CIMP-low phenotype (42.1%; *p* < 0.001) (expected value is 27.4%, according to published series). Interestingly, cluster 3 stage II CC patients showed a higher than expected proportion of dMMR (35.6%), *BRAF* mutated (24.2%), and CIMP-High cases (39.3%). Of note, of 32 cluster 3 patients with *BRAF*-mutation, 24 were also dMMR and CIMP-High (Table 2). Cluster 3 stage II CC also showed an increased proportion of women (53%) and right colon tumour localization (59.5%). Finally, cluster 4 patients showed a lower incidence of *KRAS* mutations (24.1%), dMMR phenotype (8.9%) and a higher proportion of no-CIMP (63.4%). Cluster 4 patients were also predominantly *BRAF* WT (94.9%). Taken together, these analyses confirmed the heterogeneity of stage II CC while uncovering potentially novel stage II CC subtypes based on DNA methylation patterns across 40 genes. To assess how these newly discovered DNA methylation clusters relate to global DNA hypomethylation profiles, we performed DNA methylation analysis of LINE-1 elements, since hypomethylation at these and other repeat elements is considered a hallmark of early CC development [20]. Not unexpectedly, all of our cases showed evidence of global LINE-1 hypomethylation (average decrease of 21.4%) compared to non-cancer, control colon tissue (Figure 2b). Interestingly, clusters 1 and 3 showed more marked LINE-1 hypomethylation than clusters 2 and 4. Taken together, these data indicate that our 40 CpG site classifier describes novel disease subtypes in stage II CC and suggest their emergence most likely proceeds global hypomethylation of LINE-1 elements, possibly in a step-wise, ‘branched’ fashion. Our next step was to replicate our findings in an independent patient series. For this, we selected 134 samples (116 tumour and 18 adjacent normal tissue samples) with available DNA methylation data, from the TCGA portal, and applied the same unsupervised hierarchical clustering strategy, as conducted in the discovery cohort, by using a set of 62 CpG sites that closely mapped to or nearby the 40 CpG sites of our initial classifier (Figure 2a, right panel and Appendix A). While cluster 1 patients were not identified in the TCGA cohort (rarest subtype in the discovery cohort), cluster 2, 3, and 4 subtypes were clearly identified, thus validating our findings from the registry-based population of stage II CC (Figure 2a). While the TCGA data did not permit further investigations of links with other clinical-molecular variables due to incomplete data, RNAseq data allowed us to perform a GSEA analysis interrogating the MSigDB hallmark gene set [21] and to examine immune landscapes by using the iAtlas algorithm [19]. Compared to normal colon tissue, cluster 2, 3, and 4 stage II CC tumours showed marked enrichment for MYC/E2F-dependent proliferation signatures (nominal *p*-value < 0.05) by GSEA analysis, without evidence of major biological differences beyond that (not shown). By iAtlas analysis, immune landscapes showed a subtle shift towards ‘IFNγ dominant’ subtype (C2) cases in clusters 3 and 4, compared to cluster 2 stage II CC (Appendix A). It is of note that ‘TGFβ dominant’ (C6) subtype cases—reported as a poor prognosis in numerous cancers [19]—were detected in the cluster 4 stage II CC patient subgroup, but not in other clusters.

### 3.2. Identification of CDH17 and LRP2 DNA Methylation Status as Independent Predictive Markers of Disease Recurrence in Stage II CC

We next asked whether our stage II CC DNA methylation-based classifier could be of prognostic value for assessment of recurrence risk in stage II CC by using our population-based cohort of 383 patients. Looking at recurrence risk, we noted only one relapse case in cluster 1 compared to other clusters (Appendix A), which together with immune feature data was suggestive of this cluster being a less aggressive entity. We then assessed the predictive value of individual CpG classifier sites, first by using univariate analysis (Appendix A) across patient subgroups, according to DNA methylation beta values grouped in quartile increments. Quite strikingly, by this approach, six CpG sites reached significance for prediction of disease recurrence by univariate analysis, of which two (CDH17_E31_F and LRP2_E20_F) showed independent predictive value in a multivariate model (Figure 2c and Appendix A). Hypermethylation of the CDH17_E31_F CpG site (Q4: 75 to 100% beta values) was associated with a significantly higher incidence (LR chi2(3) = 14.26, Prob > chi2 = 0.0026) of tumour recurrence (Figure 2c, upper panels) compared to the three other quartiles. *CDH17* encodes Cadherin 17 (also known as liver intestine or LI-cadherin), which functions in cell–cell adhesion and which shows deregulated expression in gastric and metastatic CC cancers [22]. By contrast, relative hypermethylation at the LRP2_E20_F CpG site (Q4; 75 to 100%) was associated with significantly lower incidence (LR chi2(3) = 17.07, *p* > chi2 = 0.0007) of tumour recurrence compared to all other quartiles (Figure 2c, lower panels). *LRP2* (also called Megalin) encodes the low density lipoprotein receptor-related protein 2, a multi-ligand endocytic receptor, which is also involved in mitochondrial homeostasis [23] and cell signalling of developmental pathways [24]. Altered *LRP2* expression and/or mutations have been described in various cancers, including CC [25,26,27].

### 3.3. CDH17 and LRP2 Expression According to DNA Methylation Status in Stage II Colon Cancer

In view of our clinical findings, the next question was to assess the potential impact of CpG methylation on the expression of *CDH17* and *LRP2*, respectively, in stage II CC. For this we took advantage of the RNAseq data for the TCGA stage II CC cohort. The CpG site of interest in the *CDH17* gene is localised 31 nucleotides downstream from the *CDH17* transcription start site (TSS), in a CpG poor region (eight CpG sites in a region of 700 base pairs), thus raising the possibility of an influence of this site on *CDH17* expression (Figure 3a). In keeping with this, by linear regression analysis, a significant inverse correlation between DNA methylation at CpG site cg17768665 at the *CDH17* gene promoter and *CDH17* expression was observed (Figure 3b; R = -0.37 and *p* = 0.00012). Additionally, *CDH17* expression was lowest in the highest quartile methylation group (Q4, teal box plot), compared to the control cases (Q1 to 3, red box plot) (Figure 3c), further supporting an active role for DNA methylation in regulation of *CDH17*. Re-inspection of the distribution of beta values for CDH17_E31_F DNA methylation levels in our discovery cohort showed these to be homogeneously distributed from 0.03 to 0.96, suggestive of a progressive methylation process during tumour progression (Figure 2c, upper left panel). The CpG site of interest for the *LRP2* gene is located 20 bases downstream of the *LRP2* TSS site in a large CpG island spanning more than 1 kb and comprising a total of 52 CpG sites (Appendix A). Hypermethylation at this site could reasonably be predicted to repress transcriptional activity of *LRP2*. *LRP2* transcripts are undetectable in normal colon tissues, consistent with hypermethylation at this site in these tissues. However, in most stage II CC tumours in the TCGA cohort studied here, decreased DNA methylation was not associated to increased transcript levels for *LRP2* (Appendix A), which is suggestive of a requirement for additional (epi)genetic mechanisms for full transcriptional derepression of *LRP2* in colon cancer.

### 3.4. Functional Enrichment and Immune Cell Subset Analyses in TCGA Stage II CC According to CDH17 and LRP2 Methylation Status

We next investigated the biological processes that might underlie the clinical prognostic significance of *CDH17* and *LRP2*, respectively, in stage II colon cancer. For this, we first performed a GSEA analysis of cases presenting hyper (Q4) versus hypo-methylated (Q1 to 3) *CDH17* or *LRP2*, respectively, taking advantage of the RNAseq data for these patients in the TCGA stage II CC cohort (Figure 4a,b). By GSEA, hyper versus hypo-methylated *CDH17* patients (methylation quartile 4; 75–100% compared to grouped methylation quartiles; 0–75%) showed differential expression of a set of 2698 genes (1959 over-expressed and 739 under-expressed genes, respectively). Subsequent hallmark MSigDB analysis revealed 42 gene sets with a positive normalised enrichment score (NES) that were differentially enriched between *CDH17* hypermethylated versus hypomethylated CC cases. Among these hallmark gene sets, 18 were considered as significant (FDR < 0.25 and *p*-value < 0.05) (Appendix A). A remarkable finding was enrichment in hallmarks relating to KRAS signalling, epithelial–mesenchymal transition, and inflammatory/immune function (interferon gamma/IL2-STAT5 hallmarks) in the *CDH17* ‘high methylation’ CC patients compared to all others (Figure 4a). To more fully characterise *CDH17* hyper versus hypo-methylated CC cases, immune cell composition was then studied by using ImmuCellAI analysis (Immune Cell Abundance Identifier) [18]. CD4, Tc, Tex (“exhausted” T cell phenotype), nTreg, Th17, MAIT, and macrophage subsets were positively correlated with high compared to low methylation of *CDH17* (classed by quartiles, as for GSEA) (Figure 4c and Appendix A). In addition, NKT, neutrophil, and CD8 T cell subsets were negatively correlated with *CDH17* ‘high methylation’ compared to ‘low methylation’ (classed by quartiles, as for GSEA) (Figure 4c and Appendix A). Taken together, these data are suggestive of defective tumour immune surveillance in *CDH17* ‘high’ versus ‘low methylation’ status stage II CC tumours, despite evidence by RNAseq of expression of *IFN-γ* and *CXCL10* (top gene in the IFN beta signature, by GSEA).

Using the same approach, GSEA analysis comparing *LRP2* CpG ‘high’ (75% to 100%; good prognosis CC patient group) versus *LRP2* CpG ‘low’ (0 to 75%) methylation group quartile patients revealed differential expression of 1072 genes (363 over-expressed and 709 under-expressed genes). Functional annotation by GSEA revealed 32 hallmark gene sets with a positive NES, of which five with significant enrichment in *LRP2* ‘high’ versus ‘low’ methylation CC patient groups (Appendix A). Quite strikingly, the enriched functions included mTORC1 and (p53-independent) DNA repair hallmarks, respectively, in addition to multiple immune signatures related to B, CD4 T cell, and TFH function (Figure 4b, Appendix A). By ImmuCellAI analysis, Tc, DC, and Tgd subsets were positively correlated to *LRP2* ‘high methylation’ status (classed by quartile, as for GSEA). Although not reaching statistical significance, the B lymphocyte subset was also enriched, in keeping with the GSEA analysis in the same patient group (*p* = 0.069). By contrast, the iTreg subset was negatively correlated with *LRP2* ‘high methylation’ status, together with a tendency towards depletion of the CD4 T lymphocyte subset, although this did not reach statistical significance (Figure 4d and Appendix A). By iAtlas analysis, an increased number of ‘IFNdominant’ immune subtype cases were seen in the *CDH17* ‘high’ versus ‘low’ methylation tumour subgroups (Figure 4e). By the same approach, ‘TGFβ-dominant’ (C6) and ‘lymphocyte depletion’(C4) immune subtypes were absent from the *LRP2* ‘high’ versus ‘low’ methylation tumour groups (Figure 4e).

We next asked how these immune cell subset distributions, particularly in the *CDH17* ‘high methylation’ compared to *CDH17* ‘low-methylation’ cases, might relate to the cytokine [*IL-12*, *IFN*, *IL-10*, *IL17*] and chemokine (*CXCL9*, *10*, and *11*, *CCL2*) ‘milieu’, to the expression of *TGFB1* (encodes TFGβ), and to granzyme B (*GZMB*) and perforin (*PRF1*) levels, which could be expected to be increased following augmented recruitment of MAIT and increased natural killer cell activation. For this analysis, we used the same RNAseq data as used for ImmuCellAI analysis to analyse transcript levels of the above cytokines/chemokines, TFGβ and enzymes in our stage II CC tumours (TCGA) (Figure 5). *CDH17* ‘high-methylation’ cases, showed increased expression of *IL12B*, *IL10*, *CXCL9* and *10* and *CCL2* compared to *CDH17* ‘low-methylation’ cases, suggestive of a favourable cytokine/chemokine ‘milieu’ in these tumours. However, this was accompanied by significantly higher *TGFB1* levels and unchanged *IL17* levels, despite evidence of increased Th17 cell recruitment in *CDH17* high versus low-methylation tumours (Figure 5). Of note also, *GZMB* transcripts (encoding Granzyme B) were similar across these two groups, consistent with the notion that cytotoxic T cell function might be compromised in *CDH17* ‘high methylation’ cases (Figure 3) and with the fact these cases showed signs of cytotoxic T cell ‘exhaustion’ (Figure 3 and Appendix A, showing increased expression levels of immune checkpoint inhibitors—*TIM-3*, *PD-1*, *PD-L1*, *CTLA4*, *LAG3*—and *IL2RB*, in these cases). By comparison, no significant differences in expression of these cytokines/chemokines or of *TGFB1*, *GZMB* or *PRF1* were seen between the *LRP2* high methylation versus low-methylation cases.

Insufficient material was available to explore these findings further in tissue samples in our discovery cohort, but immunohistochemistry in a subset of *CDH17* ‘high methylation’ (*n* = 11) versus ‘low methylation’ (*n* = 34) cases showed globally similar infiltration by T cells (CD3). This was also the case for *LRP2* ‘high’ (*n* = 8) vs. ‘low-methylation’ (*n* = 37) cases (Appendix A).

## 4. Discussion

By our biomarker discovery approach, we have identified a novel 40-gene CpG site classifier that could separate stage II CC into four discreet clusters, which partially, but not completely, overlap with known molecular features of CC (dMMR, CIMP-H, *BRAF* mutation, for example), thus suggestive of novel molecular substructure among stage II CC cases, at least regarding epigenetic control by DNA methylation. By examining LINE-1 methylation status in the same patients, we provide evidence that our DNA-methylation-based stage II CC clusters arise subsequent to global LINE-1 hypomethylation, possibly as a consequence of step wise, Darwinian selection processes.

By multivariate survival analyses we then successfully achieved our goals of identifying novel predictive biomarkers of recurrence risk in stage II CC. First, we identified hypermethylation of *CDH17* as a predictor of high risk of recurrence in these patients, suggestive of a role for this factor as a tumour suppressor in the early stage CC. Cadherin-17 (also known as liver–intestine cadherin or LI-cadherin), together with Cadherin-16, belongs to the 7D cadherin family, characterised by having seven extracellular (EC) domains. *CDH17* is expressed in epithelial cells of the intestine, kidney, and developing brain, as well as in memory B cells and is misregulated in numerous cancers [22]. In a Cdh17 knockout (KO) mouse model, loss of cadherin-17 has been shown to result in increased permeability and susceptibility to chemically induced colitis and to enhanced tumour formation and progression [28]. *CDH17* also appears to control apoptotic responses to the death receptor TRAIL in CC cell lines and xenotransplant assays [29]. Here, GSEA analysis of high risk hypermethylated *CDH17* stage II CC tumours versus control cases (TCGA cohort) revealed enrichment of signatures related to RAS activation, EMT, and immune/inflammatory signalling, known to be important facets of CC biology, as evidenced by their presence in CMS (subtypes 3 and 4, for example) [9] and in mouse models of microbiota-dependent colon carcinoma [30]. Interestingly, by a different strategy, EMT-related signatures have been implicated in high-risk stage II CC [31]. Whether this relates to ‘true’ EMT (cellular plasticity) or reflects stromal/immune cell enrichment remains to be determined.

In addition, we provide evidence that stage II CC, characterised by *CDH17* ‘high methylation’, shows signs of immune cell evasion, characterised, in particular, by reduced cytotoxic T cells and features of T cell exhaustion (including overexpression of checkpoint inhibitors, such as PD-1 and PD-L1, for example), despite an apparent favourable ‘cytokine/chemokine’ milieu and seemingly robust recruitment of numerous immune effectors (cytotoxic CD8 T cells, Th17 and MAIT cells and also macrophages), except for NKT cells, which appear under-represented/excluded in these tumours compared to *CDH17* ‘low methylation’ tumours (despite increased *IL12* expression, for example). It is also worth noting that *IL17* expression levels remain unchanged in *CDH17* ‘high’ versus ‘low-methylation’ cases, despite apparent increased recruitment of *IL17*-producing immune cell types (Th17, for example), in the former. This might lead to suboptimal killing by IL-17-mediated mechanisms in these tumours. Finally, *CDH17* ‘high-methylation’ tumours display significantly increased *TGFB1* expression when compared to *CDH17* ‘low-methylation’ CC tumours. TGFβ is implicated in complex pro-tumoral, immune-stromal microenvironment remodeling, ultimately leading to immune-suppression and an altered extra-cellular matrix, notably via cancer-associated-fibroblast pro-tumoral signaling [19]. This occurs in addition to consequences on the tumour cell phenotype itself, notably induction of EMT [19], which interestingly is also a characteristic of the *CDH17*-high methylation CC cases in our study. Taken together, these data indicate that *CDH17* ‘high-methylation’ stage II CC defines an immunologically ‘cold’ tumour subset within stage II CC. Additional mechanistic studies are required to unravel the complex nature of the immune dysfunction observed. Avenues of investigation include exploring the role of pro-tumoral IFNγ and TGFβ signalling (immune-suppressive), which are predicted to globally impact the recruitment, expansion, and function of immune cells in the tumour microenvironment. This, together with probable suboptimal cytokine/chemokine responses in the immune cells themselves, such as recently shown for *CXCL10*, identified by us and others as a key cytokine for cytotoxic T cell recruitment and killing activity in solid cancers [32], could also play a role. It is also likely that immune checkpoint inhibitor function contributes, at least in part, to the suboptimal anti-tumour immunity in *CDH17* ‘high methylation’ tumours and that immune checkpoint blockade could be of interest to alleviate this. Finally, it is possible that the tumour cells themselves exhibit lowered immunogenicity and immunosuppressive functions, which might further impact expansion and/or function of certain immune cell subsets.

A second factor, LRP2, for which gene hypermethylation showed prognostic value for identification of a subgroup of stage II CC patients of particularly low risk of recurrence, was also identified by our biomarker discovery pipeline. LRP2 plays a role as a multi-ligand endocytic receptor of plasma proteins, vitamins, and hormones in a cell context dependent manner. Interestingly, LRP2 is also implicated in mitochondrial homeostasis [23] and has been shown to function as an auxiliary receptor for developmental signalling via the SHH (Sonic Hedgehog) in neuro-epithelial tissues [24]. Misregulated expression and mutations of *LRP2* are described in CC, but functional and clinical significance are unknown [25,27]. Our data suggest that *LRP2* ‘high methylation’ stage II CC cases comprise a distinct clinical and biological entity with very low risk of recurrence after surgery and display hallmarks related to mTORC1 and DNA repair signalling, together with immune signatures related to B cell immunity (GSEA analysis) and dendritic cell function. Enrichment for B cell functions in our low risk stage II CC patients evokes a role for B cells in immune defence in stage II CC, possibly within tertiary lymphoid structures, as seen in other cancers [33] and/or via specific B cell regulatory functions that have been shown to protect against inflammation-associated tumorigenesis in mouse models of colitis [34].

Further investigations in larger prospective series are planned to explore our findings relating to recurrence risk and disease (immune) biology in stage II CC patients. First, prospective evaluation of the prognostic value of our DNA methylation classifier is planned in patients enrolled in the PRODIGE 13 study (NCT00995202) [35]. Second, an ancillary study testing the prognostic value of *CDH17* and *LRP2* methylation profiling is planned in patients enrolled in the PRODIGE 70-CIRCULATE randomised study [EudraCT no. 2019-000935-15; [36]; test adjuvant chemotherapy is guided by ctDNA]. Since the DNA methylation events described in our CC patient series track with LINE-1 hypomethylation, it will be of interest to establish the potential role of these elements, and their transcriptional activation status in the observed disease features in stage II CC [37].

## 5. Conclusions

In summary, our biomarker discovery strategy has shown to be a powerful means to identify clinically actionable predictive biomarkers of disease recurrence in stage II CC, with potential for prospective validation by targeted assays in the future.

## Figures and Tables

**Figure 1 cancers-15-00158-f001:**
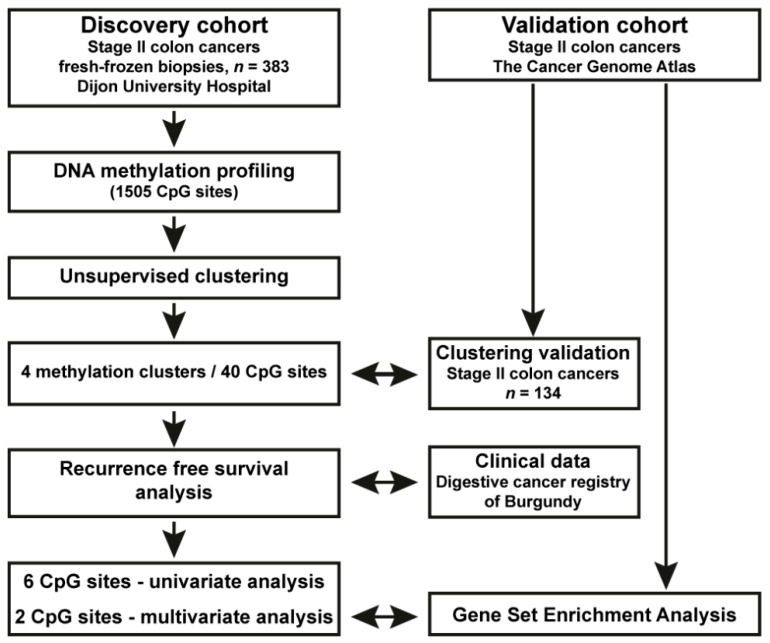
Study design. Scheme of the study design and key results. A discovery cohort composed of 383 fresh-frozen biopsies, from stage II colon cancer patients treated at Dijon University Hospital and registered in the Burgundy digestive cancer registry were selected for DNA methylation screening followed by unsupervised clustering analysis to uncover novel disease entities and DNA-methylation-based biomarkers for disease recurrence. By this strategy, we identified 40 CpG sites that could classify stage II CC into four disease clusters. By recurrence free survival analysis, two CpG sites in the *CDH17* and *LRP2* genes, respectively, were found to be significantly associated with probability of disease recurrence. A validation cohort was defined using stage II CC patient data from the Cancer Genome Atlas project (TCGA) (DNA methylation array and RNA-seq) (*n* = 134; 116 tumour and 18 adjacent normal tissue samples). By this approach, the CpG classifier could be independently validated, and functional and immune cell analyses could be performed, according to the methylation status of the 2 CpG sites of interest at the *CDH17* and *LRP2*, respectively (see text for details).

**Figure 2 cancers-15-00158-f002:**
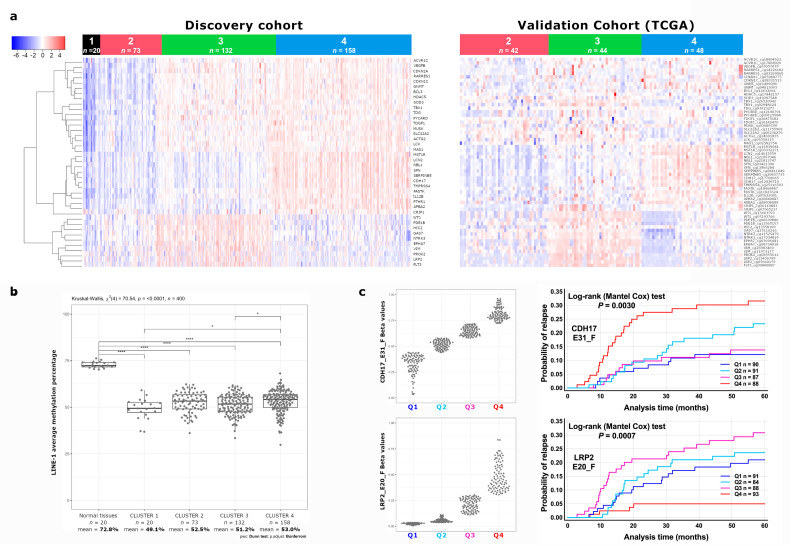
Integrated clinical and DNA methylation analysis identifies a 40 CpG site-based classifier comprising two CpG sites at the *CDH17* and *LRP2* gene promoters that independently predict risk recurrence in stage II CC. a, Left panel, heatmap showing the four methylation clusters identified from the discovery cohort, using a set of 40 CpG sites (MethylColon II classifier). (**a**) Right panel, heatmap showing the three methylation clusters identified from the validation cohort (TCGA) using a set of 62 equivalent CpG sites. (**b**) Scatter plot of average LINE-1 DNA methylation status (%) in 20 adjacent normal colon tissues and according to methylation clusters of the stage II CC discovery cohort. A boxplot is represented for each group. The result of a Kruskal-Wallis test, indicated in the above panel (**b**), showed a significant difference in methylation percentages between groups (*p*-value < 0.0001). A Dunn test was performed, and the significance level of *p*-values between each group is indicated by asterisks (* *p*-value ≤ 0.05, **** *p*-value ≤ 0.0001). Significant differences of LINE-1 methylation percentage between CpG clusters are indicated by an asterisk. An average decrease of 21.4% of DNA methylation was observed between normal tissues and stage II colon tumours (all clusters confounded). (**c**) Left panel, scatter plots of beta values for the CDH17_E31_F (upper left panel) and LRP2_E20_F CpG sites, respectively, (lower left panel), by quartile groups, as indicated. Right panel, cumulative probability of relapse for each of four patient groups defined by methylation levels, according to quartiles, for CDH17_E31_F (upper right panel) and LRP2_E20_F CpG sites (bottom right panel), respectively. The *p*-value of the log-rank test is indicated in each graph.

**Figure 3 cancers-15-00158-f003:**
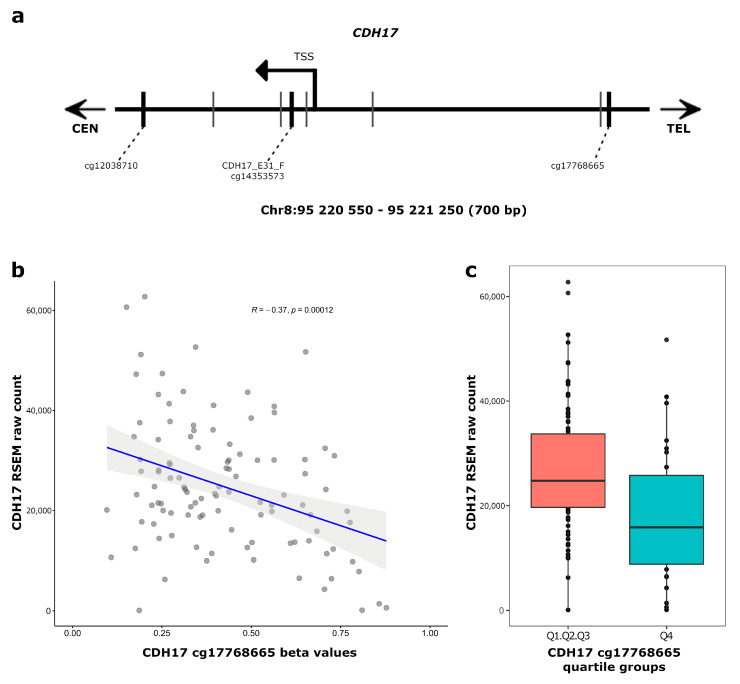
Correlation between DNA methylation and gene expression for *CDH17*. (**a**) Map of the localization of the CpG sites in the DNA region Chr8:95220550-95221250 for the *CDH17* gene. CpG sites cited in the study are represented by thick, black vertical lines and other CpG sites are represented by thin grey vertical lines. (**b**) Linear regression plot between the RNAseq expression level of *CDH17* (RSEM raw count) and the methylation level (beta value) at the cg17768665 CpG site using data from the TCGA consortium for 108 stage II patients. The Pearson’s coefficient correlation (R) was −0.37, and the *p*-value (*p*) was 0.00012 (indicated in the top of the plot). (**c**) Scatter plot of the RNAseq expression levels for *CDH17* (RSEM raw count) for quartile groups Q1/Q2/Q3 (the cases corresponding of 75% lowest beta values for the cg17768665 CpG site) vs. quartile group 4 (the cases corresponding of the 25% highest beta values). A boxplot is represented for each group.

**Figure 4 cancers-15-00158-f004:**
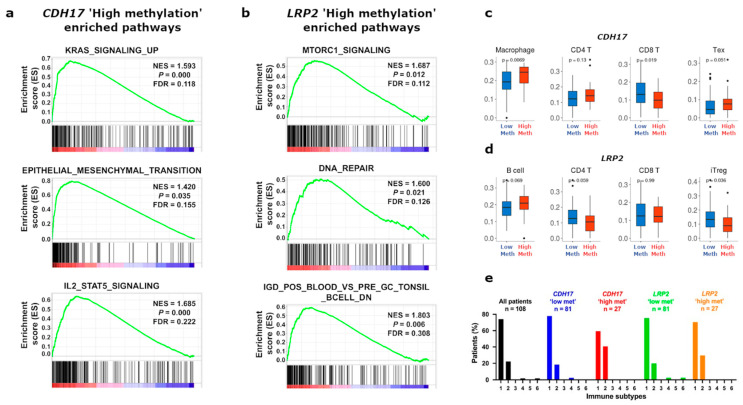
Functional and immune signatures according to *CDH17* and *LRP2* methylation status in stage II CC (TCGA). (**a**) Enrichment plots from GSEA analysis, according to *CDH17* methylation status, showing three hallmark gene sets (H) significantly enriched in the hypermethylated *CDH17* stage II CC tumours, as follows: IL2_STAT5_SIGNALING, KRAS_SIGNALING_UP and EPITHELIAL_MESENCHYMAL_TRANSITION (normalised enrichment score (NES) of 1.69, 1.59 and 1.42, a *p*-value of 0.00, 0.00 and 0.04 and a false discovery rate (FDR) of 0.22, 0.12 and 0.16, respectively: Q4 hypermethylated *CDH17* cases; left side of GSEA plots, as indicated). (**b**) Enrichment plots from GSEA analysis, according to *LRP2* methylation status, shows two hallmark gene sets (H) and one immunologic signature gene set (C7) that are significantly enriched in *LRP2* high methylation stage II CC cases compared to all other cases, as follows: DNA_REPAIR, MTORC1_SIGNALING and GSE12845_IGD_POS_BLOOD_VS_PRE_GC_TONSIL_BCELL_DN with NES scores, respectively, of 1.60, 1.69 and 1.80, *p*-values of 0.02, 0.01 and 0.01 and FDR of 0.13, 0.11 and 0.31. Q4 hypermethylated *LRP2* cases; left side of GSEA plots, as indicated. (**c**) Box plots from ImmuCellAI analysis showing immune cell subset abundance between *CDH17* ‘Low methylation’ (quartile groups Q1/Q2/Q3) versus *CDH17* ‘High methylation’ (Q4) subgroups, respectively (*p*-values, as indicated). (**d**) Box plots from ImmuCellAI analysis showing immune cell subset abundance between *LRP2* ‘Low methylation’ (quartiles groups Q1/Q2/Q3) and *LRP2* ‘High methylation’ (Q4), respectively (*p*-values, as indicated). (**e**) Bar charts showing immune landscapes by iAtlas analysis: percentage of CC tumours classified according to six different immune subtypes (numbered 1 to 6 on x-axis: 1, Wound Healing; 2, IFNγ-Dominant; 3, Inflammatory; 4, Lymphocyte Depleted; 5, Immunologically Quiet; 6, TGFβ-Dominant (see text for details)).

**Figure 5 cancers-15-00158-f005:**
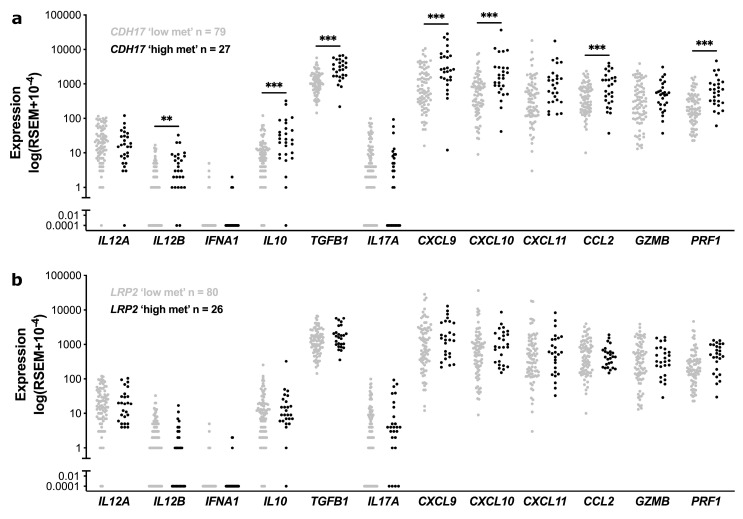
Expression of cytokines related to CD8 T cells (Tc1), NKT, Th17 and MAIT cells, *TGFB1*, *GZMB,* and *PRF1* genes according to *CDH17* and *LRP2* methylation status in stage II CC (TCGA). (**A**) Scatter plot of RNAseq expression levels (log-transformed RSEM raw count) between *CDH17* ‘low methylation’ (quartile groups Q1/Q2/Q3, grey dots) versus *CDH17* ‘high methylation’ (Q4, black dots) subgroups, as indicated. An unpaired Student’s *t*-Test was performed; the significance level of p-values between groups are indicated by asterisks (** *p*-value ≤ 0.01, *** *p*-value ≤ 0.001). (**B**) Scatter plot of RNAseq expression levels between *LRP2* ‘low methylation’ (quartile groups Q1/Q2/Q3, grey dots) versus *LRP2* ‘high methylation’ (Q4, black dots) subgroups, as indicated. No significant differences by the Student’s *t*-Test were observed for the indicated cytokines, *TGFB1*, *GZMB,* or *PRF1*.

**Table 1 cancers-15-00158-t001:** Clinical and molecular characteristics of the stage II colon cancer population-based discovery cohort.

A.				B.			
		*N* = 383				*N* = 383	
		*n*	%			*n*	%
	**Sex**				**MSI status**		
	Women	169	44.1		MSI	85	22.2
	Men	214	55.9		MSS	298	77.8
	**Age class**				***KRAS* codon 12-13 mutational status**		
	≤64 years old	87	22.7		Mutated	116	30.3
	65–74 years old	105	27.4		Wild-type	266	69.5
	≥75 years old	191	49.9		Unavailable	1	
	**Tumour site**				***BRAF* codon 600 mutational status**		
	right colon	172	44.9		Mutated	54	14.1
	left colon	141	36.8		Wild-type	329	85.9
	recto-sigmoïd junction	65	17.0		***PIK3CA* codon 542 and 1047 mutational status**		
	Unavailable	5			Mutated	70	18.3
					Wild-type	312	81.5
					Unavailable	1	
	**Staging**				**CIMP status**		
	T3	97	25.3		CIMP-High	66	17.2
	T4	260	67.9		CIMP-Low	105	27.4
	locoregional extension	26	6.8		No CIMP	144	37.6
					Unavailable	68	17.8
	**Chemotherapy treatment**				***ERBB2* CNV status**		
	yes	46	12.0		Amplification	42	11.0
	no	337	88.0		No amplification	274	71.5
					Unavailable	67	17.5
	**Five-year followup reccurence**				***TP53* CNV status**		
	no	315	81.8		Deletion	80	20.9
	yes	68	17.7		No Deletion	236	61.6
					Unavailable	67	17.5

**Table 2 cancers-15-00158-t002:** Clinical and molecular associations with DNA methylation clusters in the population-based discovery cohort of stage II CC.

		**Cluster 2**		**Cluster 3**		**Cluster 4**		
		(N = 73)		(N = 132)		(N = 158)		Khi2 Test
	N	n	%	n	%	n	%	*p*
**Sex**								
Men	214	36	49.3	62	47.0	105	**66.5**	**0.002**
Women	169	37	50.7	70	**53.0**	53	33.5	
**Tumour site**								
right colon	172	32	45.7	78	**59.5**	52	33.1	**<0.001**
left colon	141	24	34.3	40	30.5	70	**44.6**	
recto-sigmoïd junction	65	14	20.0	13	9.9	35	22.3	
** *KRAS* **								
Mutation	116	30	41.1	45	34.4	38	24.1	0.021
Wild -type	266	43	58.9	86	65.6	120	75.9	
**MSI phenotype**								
MSI	85	19	26.0	47	**35.6**	14	8.9	**<0.001**
MSS	298	54	74.0	85	64.4	144	**91.1**	
**CIMP phenotype**								
CIMP-High	66	11	19.3	42	**39.3**	9	6.9	**<0.001**
CIMP-Low	105	24	**42.1**	36	33.6	39	29.8	
No-CIMP	144	22	38.6	29	27.1	83	**63.4**	
** *BRAF* **								
Mutation	54	12	16.4	32	**24.2**	8	5.1	**<0.001**
Wild -type	329	61	83.6	100	75.8	150	**94.9**	
**MSI/*BRAF* status**								
MSI/BRAF Mutation	46	11	15.1	28	**21.2**	5	3.2	**<0.001**
Others	317	62	84.9	104	78.8	153	**96.8**	
**MSI/CIMP/*BRAF* status**								
MSI/CIMP-High/BRAF Mutation	38	9	12.3	24	**18.2**	3	1.9	**<0.001**
Others	325	64	87.7	108	81.8	155	**98.1**	
***ERBB2* copy number**								
Amplification	42	9	15.5	12	11.3	19	14.3	0.703
Deletion/Neutral	274	49	84.5	94	88.7	114	85.7	
***TP53* copy number**								
Deletion	80	9	15.5	33	31.1	31	23.3	0.077
Amplification/Neutral	236	49	84.5	73	68.9	102	76.7	

This table describes the number and the corresponding proportion (%) of patients within each DNA methylation cluster, according to clinical and molecular characteristics, as indicated. The *p*-value of a Chi2 test is shown for each association (color-coded per cluster, as indicated, for data comparisons that reach significance; see text for details). MSI (microsatellite instability) equivalent to dMMR (deficient mismatch repair).

## Data Availability

The data generated in this study are not publicly available, as they contain information that could compromise patient consent, but are available upon reasonable request from the corresponding authors.

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
