# Peer review of "Integrative Clinical and DNA Methylation Analyses in a Population-Based Cohort Identifies CDH17 and LRP2 as Risk Recurrence Factors in Stage II Colon Cancer"

_cancers, 2022, doi:10.3390/cancers15010158_

Round 1

Reviewer 1 Report (Previous Reviewer 1)

General comments: The authors haves significantly  improved the quality of the manuscript. Thus, the current version answered all my concerns and it is ready for publication.

This manuscript is a resubmission of an earlier submission. The following is a list of the peer review reports and author responses from that submission.

Round 1

Reviewer 1 Report

General comments: The authors performed a bulk RNA sequence, DNA methylation-based analysis and clinical screen in tumor and adjacent normal mucosa samples collected from 383 patients with stage II colon cancer and nine DNA samples from normal colon mucosa. The authors found a unique gene signature that classifies colon cancer into four subtypes. They also found that hypermethylation of the CDH17 gene predicts high cancer recurrence associated with activation of KRAS signaling pathway and a potential decrease in anti-tumor immunity. In contrast, hypermethylation of the LRP2 gene is associated with a probable enhance in tumor immunity and DNA repair pathways.

Specific comments:

1.     Figures 2 and 3 are not legible. The colors selected for each component of the graphs are not adequate and the author should avoid to use bold font and shadow. 

2.     The authors must provide the meaning of *, **, ** and add the p-value in the figure legends 2 and 3.

3.     The authors found increased MAIT cells (supplementary Figure 4a), decreased NKT cells, neutrophils  in  CDH17 high compared to low methylation cases.  However, the increase in MAIT cells in tumors correlate with high expression  of granzyme B, perforin and increased natural killer cells activation. In addition, Th17 cells induce tumor killing by IL-17-mediated mechanism. Could the authors add 1) a new analysis showing cytokines (IL-12, IFN, IL-10 and TGF beta, IL17), chemokines (CXCL9, 10, and 11, CCL2), granzyme B and perforin produced by type 1 cytotoxic CD8 T cells (Tc1), NKT, Th17 and MAIT cells;. 2) A possible explanation about the increase of MAIT cell and decrease in natural killer cells in high methylation of CDH17 vs. LRP2  vs molecules related with tumor.

4.     The author might validate the gene analysis expression on tissue sections from hypermethylated CDH17 and LRP2 stage II cancer colon samples. For example, they can stain using specific markers for cytotoxic CD8 T (granzyme B) and Tex cells (IL2Rb). Then, calculate the ratio Tc/Tex and correlate it with the methylation data set.

5.     The author must check the manuscript carefully to correct several typos thought the text and missing spaces. 

Reviewer 2 Report

This study describes a methodology to integrate clinical information with DNA methylation analysis, and  this strategy is employed to successfully identify the hypermethylation of CDH17 and LRP2 to be the predictive biomarkers of recurrence of stage II colon cancer. This study is well-written with comprehensive data analysis and logical thinking. The large cohort dataset further solidates the discoveries and conclusions. Overall, the paper is in good shape.

 1. The authors could clarify whether the DNA extraction and methylation analysis were executed internally or externally.  More details of the experiment part could be added to Part 2.

2. The quality of Figure 2 and Figure 3 can be improved. The current version is blurry and hard to read.
